# Assessment and Feedback Control of Paving Quality of Earth-Rock Dam Based on OODA Loop

**DOI:** 10.3390/s21227756

**Published:** 2021-11-22

**Authors:** Cheng Wang, Jiajun Wang, Wenlong Chen, Jia Yu, Zheng Jiao, Hongling Yu

**Affiliations:** State Key Laboratory of Hydraulic Engineering Simulation and Safety, Tianjin University, Tianjin 300350, China; chengwtju@tju.edu.cn (C.W.); chenwenlong@tju.edu.cn (W.C.); yujia@tju.edu.cn (J.Y.); jiaozheng@tju.edu.cn (Z.J.); yuhongling@tju.edu.cn (H.Y.)

**Keywords:** paving, quality control, quality assessment, OODA loop, cellular learning automaton, dynamic path planning, GNSS

## Abstract

Paving thickness and evenness are two key factors that affect the paving operation quality of earth-rock dams. However, in the recent study, both of the key factors characterising the paving quality were measured using finite point random sampling, which resulted in subjectivity in the detection and a lag in the feedback control. At the same time, the on-site control of the paving operation quality based on experience results in a poor and unreliable paving quality. To address the above issues, in this study, a novel assessment and feedback control framework for the paving operation quality based on the observe–orient–decide–act (OODA) loop is presented. First, in the observation module, a cellular automaton is used to convert the location of the bulldozer obtained by monitoring devices into the paving thickness of the levelling layer. Second, in the orient module, the learning automaton is used to update the state of the corresponding and surrounding cells. Third, in the decision module, an overall path planning method is developed to realise feedback control of the paving thickness and evenness. Finally, in the act module, the paving thickness and evenness of the entire work unit are calculated and compared to their control thresholds to determine whether to proceed with the next OODA loop. The experiments show that the proposed method can maintain the paving thickness less than the designed standard value and effectively prevent the occurrence of ultra-thick or ultra-thin phenomena. Furthermore, the paving evenness is improved by 21.5% as compared to that obtained with the conventional paving quality control method. The framework of the paving quality assessment and feedback control proposed in this paper has extensive popularisation and application value for the same paving construction scene.

## 1. Introduction

The paving operation is an important part of the construction process of earth-rock dams [1]. Generally, bulldozers and dump trucks constitute the construction machinery for the paving of earth-rock dams for meeting the tight construction schedule. The paving work involves a series of varied operations such as cutting, carrying, spreading, and simple grading. In each stage, there are specific requirements for the operation mode, driving path, and paving parameters of bulldozers and dump trucks. Failure to meet these requirements may result in unqualified paving results such as uneven paving, thus affecting the compactness quality of the final earth-rock dam and reducing the service life of the dam [2,3,4]. Therefore, the quality evaluation and feedback control of the bulldozer paving operation is of great significance for ensuring the paving quality [5].

Levelling is a process of spreading soil or earth-rock materials evenly in the work area. Effective levelling quality control plays a crucial role in increasing the productivity of earth-rock dam construction [5]. At existing earth-rock dam construction sites, the bulldozer is used for levelling work, which frequently comprises a series of operations such as cutting, carrying, spreading, and simple grading [6]. A typical levelling method is presented in Figure 1. The compacted layer is followed by a levelling layer. Dump trucks transport earth-rock materials from stockyards over the edge of the levelling area. The bulldozer operator then performs spreading by continuously changing the blade position while moving in order to complete the levelling work. This study is focused on such levelling work.

At present, in practice, the quality control of paving is mainly dependent on traditional methods such as on-site monitoring of the paving process and random sampling after paving, which is time-consuming and laborious and provides unreliable results and limited coverage. When the levelling layer is too thick, there is a serious separation of coarse and fine materials, which results in the unstable and unsafe operation of earth-rock dams [3,4]. Furthermore, in the majority of cases, the operation path that allows bulldozers to cut and move earth-rock materials is planned based on the operators’ work experience. Most importantly, conventional levelling quality control is mainly dependent on the random spot test that is conducted after the levelling work is complete [4]. Thus, there are many problems associated with this levelling quality control method, such as delayed feedback, low productivity, and high dependence on human experience.

In this paper, we propose a novel assessment and feedback control framework for improving the levelling quality of earth-rock dams. The OODA loop, invented by Boyd and widely used in military operations, is the foundation of our framework [7]. The OODA loop is a closed-loop tactical concept that comprises the use of information interaction to optimise tactics in real time and includes observing, orienting, deciding, and acting [8]. By specifying the composition of each model of the OODA loop, we demonstrate that, in addition to being a very simple framework, it can be used to comprehensively and effectively evaluate the levelling quality during the construction process and provide timely guidance for levelling quality control.

The contributions of this paper are the following:

(1) The OODA loop is used to build a feedback framework that is applied to the paving process through the improvement of each module to realise dynamic assessment and control during the paving process. (2) A cellular learning automaton (CLA) model that improves the observe module to decode the partition, store paving information, and perceive real-time information during the paving process during the actual paving process is established. (3) A dynamic path-planning method that is embedded in the decision module of the OODA framework to guide the bulldozer is proposed for optimising the paving quality indexes. (4) The proposed methodology exhibits excellent performance in the paving of earth-rock dam construction.

The rest of this paper is organised as follows. A literature review of related works is presented in Section 2. The overall research framework of the study is presented in Section 3. In Section 4, a detailed discussion of the proposed methodology is presented. A case study of the assessment and feedback control of levelling work is presented in Section 5. This section also presents a research discussion. Finally, the conclusions and future work of this study are presented in Section 6.

## 2. Related Work

At present, in the field of automatic control systems for bulldozer equipment, with the progress of wireless communication technology, feedback control, and data analysis technology, a computer physics system has been developed, which provides a new means of controlling automated vehicles to improve production efficiency and equipment life. An increasing number of related studies are aimed at developing a powerful method for facilitating the development of autonomous bulldozer systems. In general, an autonomous navigation system should comprise terrain mapping, path planning, and navigation [9].

Navigation relies on the advanced positioning technology of bulldozers. In recent years, Chang [10] adopted a global navigation satellite system (GNSS), accelerometer, and thermal scanner installed on the roller and paver measurement instruments for collecting the construction-process information of location, thickness, temperature and other information to achieve the quality control of asphalt concrete paving. With the help of these measurement instruments, Chang reduced the influence of artificial factors and improved the accuracy of the paving quality control. Yu [11] adopted the simultaneous paving technology in the construction process of the Hong Kong–Zhuhai–Macao Bridge and improved the efficiency of the paving construction and the precision of quality control of paving by improving the construction method. In the field of gravelly soil paving, Zhong [3] proposed a real-time method for monitoring the paving construction of core earth-rock dams, realised the real-time monitoring of the bulldozer position, and generated a graphic report that reflected the construction quality after the paving. Zhong [4] used the k-nearest neighbour algorithm to solve the thickness estimation problem of the uncontacted area of a bulldozer in the paving process and, using the obtained mechanical equipment positioning information, realised the thickness monitoring of an entire warehouse surface. However, the existing research on the paving process is primarily focused on the asphalt paving field, with more attention on quality control in the construction process and the use of advanced measurement devices, improved construction technology, visual management, and other methods for achieving high-precision in-process control.

However, gravelly soil paving remains at the level of construction parameter monitoring, and still relies on manual experience for the bulldozer operation path selection, which fails to provide effective control of the paving quality. Terrain mapping provides a solution to the perception problem of real-time terrain information for bulldozers, and Komatsu uses the method of taking into consideration motion compensation and polygon approximation for updating the terrain [9].

However, all mechanical actions in the paving process are construction-quality oriented, and merely relying on the monitoring of machinery is insufficient for ensuring the quality of the entire warehouse. Cellular automata (CAs) have the characteristics of regular state transitions and discrete update storage. In addition, precise real-time positions and grid partitions are essential for calculating and storing lift thickness information [12] CLA, which is a hybrid of cellular automata (CA) and learning automata (LA), and outperforms CA in terms of learning ability, making it more environment adaptable [13]. CLA is also preferable to LA because significant amounts of CA can induce more complicated phenomena than LA [14]. CLA integrates the benefits of CA in that the phenomena are mimicked by intelligible behavioural norms mixed with the learning capabilities of LA, rather than coupled functions [15], making computer simulations easier to execute. Since the introduction of CLA [16], this model and its extensions have proven their performance in solving problems, such as gas diffusion [17], land use evolution [18], urban development [19] and flood evolution [20] community detection [21], wireless sensor network [22], resource allocation [23], and emergency evacuation management for updating pedestrian location status information updates. So far, CLA has been mainly used to solve optimisation problems and has rarely been used to model the self-organising behaviour of automated equipment [24]. The action of mechanical equipment such as bulldozers can be transformed into quality information such as the thickness and flatness of the entire warehouse surface using the advantages of CLA for state transfer rule conversion, which provides a reliable information source for the path planning of a bulldozer.

At present, in the field of path planning, global–local path planning has attracted the attention of many researchers because it can be used to determine the approximate path by comprehensively considering the conditions of the entire field via global path planning and handle unexpected situations (such as dynamic obstacles) in the process of path planning using local path planning. Wang [25] established a hybrid path planning scheme with a global–local structure while considering the dynamic constraints of an autonomous surface vehicle (ASV) using global path planning to generate optimal sparse waypoints and using local path planning at each waypoint to control the ASV in order to avoid obstacles and move to the next waypoint. Osmankovic [26] designed a multi-stage technique for dealing with path planning problems in the case of poorly traversable and partially unknown rough terrains based on the fast D* lite algorithm for global path cost-to-go computation while employing the model predictive control (MPC) planning paradigm for solving constrained optimal control problems for the purpose of local planning. Among the global path planning algorithms, the popular algorithms mainly include the rapidly-exploring random tree algorithm (RRT), artificial potential field method, and A* algorithm.

Each algorithm has different characteristics and is suitable for different types of path planning. The A* algorithm is widely used because of its fast calculation speed and optimal path trajectory length. However, the local path-planning algorithm focuses more attention on the processing of the operation processes in a small range [27]. Shi [28] proposed a rolling motion model that meets the construction principles of the lap joint method and alternate distance method for the path planning of sub-operation surfaces to realise local operation path planning. In summary, the hybrid path planning method can comprehensively consider the global trend of path planning and local dynamic adjustment for achieving good results.

It is thus apparent that it is necessary to establish a method for coupling dynamic assessment and control for gravelly soil paving. The following three issues are required to be addressed: (1) a method of improving the timeliness of the quality assessment is required to be developed and used to guide the paving process [29]; (2) a method of perceiving the paving quality information in the work area in real time during the construction process is required to be developed [2]; and (3) the influence of artificial factors should be eliminated, and stable control of the paving quality should be realised.

To address the aforementioned problem (1), the OODA (observe–orient–decide–act) loop fits well with the idea of dynamic assessment and control in this process; it was invented by Boyd and used in the field of military operations [6]. This concept only distinguishes and sorts the modules in the process but does not specify the composition of each module in detail [30]. The OODA loop is used to build a feedback framework that is applied in the paving process, through the improvement of each module, to realise dynamic assessment and control during the paving process [31]. To address problem (2), it is necessary to divide the entire work area into grids and record the position and thickness of each grid. However, the thickness of the corresponding position must be updated according to the movement of the machine. CLAs have the characteristics of partition analysis, partition storage, and corresponding updates, which coincide with the aforementioned requirements [24]. A CLA model that describes the actual paving process has been established [20,32]. A CA is used to improve the observation module to decode the partition, store paving information, and perceive real-time information during the paving process [33,34]. The learning automaton (LA) is used to improve the orientation module to analyse the relative position of the machine [35,36]. To solve problem (3), the experience of the bulldozer operator must be categorised as mechanical operation experience and thinking experience. However, differences in the mechanical operation experience have little effect on the paving quality, and thus, the method of planning the operation path of the bulldozer is of greater significance. By summarising the experience of skilled manipulators in an abstract manner in combination with the on-site construction method, a dynamic path-planning method is proposed for optimising the paving quality indexes. This method is embedded in the decision module of the OODA framework for guiding the bulldozer. In this manner, the intelligent guidance method can be used to realise the feedback control of the paving quality [37]. The practical application of a 300 m high earth-rock dam in Southwest China validates the effectiveness and superiority of the proposed framework.

## 3. Research Framework

In this study, a dynamic quality assessment and control of paving gravelly soil under the OODA framework coupled with a CLA is proposed, which includes two parts: the dynamic quality assessment and control and the engineering application.

The dynamic quality assessment and control part mainly refers to the OODA framework coupled with the CLA. The OODA framework contains four modules: observe, orient, decide, and act. In the observation module, a CA is used to sense the information of the entire paving area in real time. In the orient module, an LA is used to interact with the environment to realise the orientation of the bulldozer. In the decision module, paving-quality goal-oriented dynamic path planning is used to determine the optimal travel path of the bulldozer. In the act module, the bulldozer is operated according to the navigation, and the current paving quality is assessed dynamically.

The engineering application part is a demonstration of the effects and characteristics of this method when applied to engineering applications. The experiments and an analysis of their results show that this method provides a superior performance as compared to manual work. The composition and application of this framework are presented in Figure 2. The OODA framework consists of four modules: observe, orient, decide, and act. In the observe module, the dozer senses information about the entire paving area in real time based on a CA. In the orient module, the LA helps the dozer to interact with the environmental information to achieve the adjustment of the dozer. In the decide module, a dynamic path planning algorithm is used to plan the optimal path for the bulldozer in order to achieve the paving quality target adjustment. In the act module, the dozer is executed by navigation and the current paving quality is dynamically evaluated.

## 4. Methodologies

### 4.1. OODA Framework Coupled with CLA

#### 4.1.1. OODA Loop

The OODA loop is a closed-loop tactical concept that uses information interaction to optimise tactics in real time and includes the stages of observe, orient, decide, and act [8], as shown in Figure 3. The observation stage mainly comprises the collection of data and information related to the problem; the orient stage is used to analyse and process the information obtained via observations and make adjustments according to the specific conditions; the decision stage is used to propose solutions and measures for addressing the problem to form a solution and is the key module for realising feedback control; the act stage is used to implement the solution. This method can be used to simulate the development and operation of agents [38].

However, OODA is only a theoretical framework and does not provide detailed provisions on the content of each module [39]. When applied to the paving process, the content details of each module must be formulated according to the characteristics of the paving process.

#### 4.1.2. Cellular Learning Automaton

The CLA comprises a CA and an LA. A CA is a system based on cells that are presented in a grid-like structure [40,41]. Each cell is a discrete individual that contains certain information and rules. Its own condition is only affected by its own state and the state of the surrounding cells [42]. The corresponding formula is as follows:(1)A~(S,T,V)

*A* represents a cell comprising rules; *T* represents its rules, which can be executed spontaneously by the cell or triggered from the outside (such as by the learning automaton); *S* represents the state information of *A*, including the coordinate position, elevation, and thickness; and *V* represents the status information of the neighbours around *A*.

The LA is an abstract model that randomly selects an action from a limited set of actions and executes it in the environment. Every behaviour selected by an individual corresponds to a change in the environment [43]. After the environment changes, individuals are encouraged to produce new behaviours, thus forming a closed loop and promoting the operation of the entire system [44]. The environment can be abstracted into an array of three quantities.
*E* = {*α*,*β*,*c*}(2)

*α* represents the input of the environment (the impact of the machine on the environment), *β* represents the output of the environment (feedback after the environment is affected), and *c* represents the probability that each element in *c* corresponds to each action. Figure 4 presents the relationship between the LA and the environment.

There is an LA in each cell of the CLA. The LA in this cell generates behaviour according to the state of the surrounding neighbour cells, and this behaviour affects the state of the neighbour cells. Thus, a cycle is formed, which promotes the continuous operation of the system [45]. On this basis, a synchronous CLA and an asynchronous CLA (ACLA) are developed. The former has a common time point for all the cells, and the calculation and update are synchronised. The latter is based on the time or specific behaviour built into each cell to activate, calculate, and update in batches [14]. As only dump trucks, bulldozers, and a small number of on-site management personnel are present during the actual construction, it can be assumed that the thickness and elevation information in the cells are only affected by the dump trucks and bulldozers. CLA can convert the real-time position information of bulldozers and the unloading information of dump trucks into the dynamic paving quality information of the storehouse surface.

The storehouse surface can be divided into a 1 m × 1 m square grid, each grid corresponds to a cellular, and the location coordinate, elevation, thickness and other information of the corresponding grid are stored in the cellular, and each state information of the cellular corresponds to its state transition rule. The transfer of each state in the cellular is carried out in the following order:

The cellular coordinates will not change after the storehouse surface division stage is obtained.The cellular activation value is determined by obtaining real-time monitoring data and the activation status information of the surrounding cells. The inactive state value is 0, when the bulldozer spatial position coordinates coincide with the cellular coordinates, the activation value is 1, when the activation value of the surrounding cell is 1, the activation state value of the cell becomes 1.1. When the spatial position coordinates of the dump truck coincide with the cellular coordinates (x,y), the activation value of the cell is changed to 2, and the activation values of the cellular coordinates (x,y−1)  and (x,y+1)  are changed to 2, thus determining that the activation values of the cells existing on the width of the mound are all updated to 2. In order to ensure that all the cells on the length of the mound are activated according to position, the activation value of cellular coordinates (x+1,y) can be changed to 2.01, and so on. The other activation value of the cell is 2+0.01n(0<n<16), then the activation value of the cells on both sides of the y direction is also changed to 2+0.01n, the activation value of the cell at the  x+1 position becomes 2+0.01(n+1).The elevation can be obtained in different ways according to the activation value of the cell, when the activation value is 1 or 2, the elevation is equal to the elevation value in the real-time monitoring data of the bulldozer or dump truck. When the activation value is 1.1, because the cell is under the bulldozer, the elevation value is equal to the elevation value of cell with activation value of 1 at this time. When the activation value is 2+0.01n(0<n<16), each cell corresponds to each position of the unloading pile. Elevation updates should be made according to Equation (3)
(3)$$H=\{\begin{array}{c} H_{old}+0.15n\;\;\;\;\;0\leq n\leq 4\\ H_{old}+0.6\;\;\;\;\;\;4<n<12\\ H_{old}+2.4-0.15n\;\;\;\;\;12\leq n\leq 16 \end{array}$$
where, H is the updated elevation of this cell, m;$\;H_{old}$ is the pre renewing elevation of this cell, m; n is the distance from the starting unloading position as shown in Figure 5, m.The thickness is generally updated according to the elevation value in the cell, and because the thickness update is the last step of the cell state update, the active state value is set to 0 after the thickness update. Thickness state transfer should satisfy the Equation (4).
(4)$$d=L_{a}-L_{b}$$
where, *d* is the thickness, m;$\;L_{a}$ is the updated elevation of the cell, m;$\;L_{b}\;$ is the elevation before updated of this cell, m.

To sum up, the cellular automata model of gravelly soil paving quality is established, the storehouse surface is divided into grids, and the real-time monitoring data are obtained by using cellular automata. Then, according to the state transition rules, the paving quality information of each location is transformed and stored in the corresponding cell. As a result, the action of construction machinery is connected with the paving quality, which provides a high-precision and real-time data source for path planning.

In addition, each time it is only affected by a few cells in the area of the machine and nearby, there is no need to update all the cells, and thus, the ACLA that is updated according to the behaviour is selected.

#### 4.1.3. OODA Framework Coupled with CLA

In the OODA framework, the ACLA mainly optimises the first two modules. Because an ACLA has the characteristics of an asynchronous update, and CA and LA have different update mechanisms and rules, they can be placed in two modules to optimise the OODA framework.

In the observation module, according to the characteristics of the CA, the work block is divided into equal-sized square grids, where each grid corresponds to a cell, and the cell stores the location coordinates, elevation, thickness, and other information of the corresponding grid [46]. As shown in Figure 6, the GNSS installed on the bulldozer records the position information in real time. After the paving of the storehouse surface is completed, the fitted elevation of the storehouse surface is obtained by extracting the paving trajectory information of the storehouse surface as shown in Figure 6a. When starting construction on a new layer, the system fits and updates the elevation map by acquiring the trajectory of the bulldozer in real time as shown in Figure 6b. Figure 6c shows the superimposition of the elevation maps of two paving thin layers. The thickness d at an arbitrary location can be obtained by calculating the height difference between the two planes, as shown in Figure 6d. In this manner, the simulated work block is established, and the cells can update their internally stored information by means of an external input (see Section 4.3 for parameter acquisition details). This completes the perception of the paving quality parameters in the observation module [47].

In the orient module, the influence of the machine is driven by the LA on the simulated work block. On receiving the discharge signal, the LA at the corresponding position is activated, and the LA then drives the update of the state of the cell in the corresponding area. On receiving the positioning data of the bulldozer, the LA at the corresponding position is activated and then drives the update of the state of the cell that contains the active LA and simultaneously drives the cell information in the surrounding area to change according to the course of the bulldozer.

### 4.2. Dynamic Assessment and Control

#### 4.2.1. Dynamic Assessment

In the two modules of the aforementioned OODA framework, quality-information perception has been implemented, and it is necessary to embed the dynamic assessment in the act module to evaluate the results. The two indicators of thickness and flatness are primarily used for quality assessment [48].

Thickness detection is used to evaluate the average thickness of the entire working block. Its formula is similar to Equation (5):(5)$$avg\_t=\frac{\sum_{i=1}^{N}t_{i}}{N}$$
where avg_t represents the average value of the paving thickness in the entire work block and is measured in meters, N represents the total number of cells in the work block, and $t_{i}$ represents the thickness of the *i*-th cell in meters. From this, the average thickness of the entire working area can be obtained. According to engineering experience, for gravelly soil, the average thickness reaches 0.26 m but does not exceed 0.30 m, which indicates that the thickness reaches the standard value [3,11,23].

The flatness can be tested after the thickness is qualified. The fatness is also defined for the entire work block, for which the formula is as follows:(6)$$F=\frac{\sum_{i=1}^{N}{\left(t_{i}-avg\_t\right)}^{2}}{N}$$
where  F represents the variance of the elevation values in all cells in the entire work area.

When the thickness is qualified and the flatness is lower than a certain value, it can be considered that the entire work block has been paved. When this condition is not met, it is considered that the paving is incomplete, and a new round of the stages of observe, orient, decide, and act is implemented under the OODA framework.

#### 4.2.2. Feedback Control

The travel of the bulldozer is controllable and affects the distribution of gravelly soil in the work block. However, the decision module is a key module that affects the trend of OODA, and the path-planning algorithm can be embedded in this module to achieve quality control. To cooperate with the real-time perception of the paving quality information in the observation module, the use of dynamic path planning is proposed for realising dynamic control [49].

At the construction site, the occupation method is generally used for paving gravelly soil. This method involves pushing the soil from a high level to a low level, such that the bulldozer is on a higher surface for a longer time. Based on the summarised work experience of skilled operators, we know that the mound closest to the bulldozer will be levelled first, and the other mounds will then be levelled [50]. Accordingly, the state of the bulldozer can be categorised as a state of work and movement. In the work state, the bulldozer flattens the mound and uses the local quality assessment method to determine the thickness and flatness of the affected area; in the move state, the bulldozer finds and approaches the nearest mound according to the distance measurement function [51]. To facilitate the state transition, the cell at the end of each mound is set as the starting cell.

The work state is the main state of the bulldozer. The bulldozer determines its direction of travel according to its own coordinates and the elevation, thickness, and other information of the surrounding cells. In combination with the method of construction in the occupation method, the following action cycle can be performed. Firstly, during the forward process, on considering the large amount of soil in the front shovel and the large engine load, the effect of adjusting the direction is poor, and it is easy for the local area to become too thick, such that the bulldozer is guided in a straight line [52]. Secondly, the bulldozer leaves the higher layer in the process of moving in a straight line, and thus, there is no soil in the front shovel. At this time, the coordinate elevation of the bulldozer changes, and it can be determined that the bulldozer should start to retreat according to this change. Then, on considering the role of the scraping surface of the front shovel during the retreating process and facilitating the subsequent forward bulldozing, the bulldozer can be guided to retreat to the thicker cell according to the thickness of the rear cell. Finally, when retreating to a thicker area, the bulldozer moves backwards in a straight line until the coordinates of the bulldozer are close to the height of the paving layer, and the bulldozer can then start the next advance of bulldozing. A bulldozing process includes multiple forward and backward operation cycles. In this process, a local quality assessment is performed by determining whether the calculated average thickness and flatness of the affected area meet the standard values. Thus, an in-process feedback control mode is developed to guide the bulldozer’s operation. Equations (5) and (6) present the thickness assessment and flatness assessment functions, respectively. When the working area of the bulldozer meets the required flatness and thickness requirements, the state of the bulldozer changes from the work to the move state.

In the move state, it should first be determined which mound’s starting cell is closest to the bulldozer, and the bulldozer should then be guided to travel to this starting cell position. According to the action model of the bulldozer, it is more inclined to move in a straight line than to turn in a large angle. Therefore, it is unreasonable to use the Euclidean distance to measure the distance between the bulldozer and the starting cell. The relative positional relationship between the starting cell and bulldozer and the course of the bulldozer should be comprehensively considered, a distance measurement function should be established, the angle measurement weight should be added on the basis of the Euclidean distance measurement, and the composite distances should then be compared to obtain the optimal solution. The specific formula for this is as follows:(7)$$d_{E}=\sqrt{{\left(x_{2}-x_{1}\right)}^{2}+{\left(y_{2}-y_{1}\right)}^{2}}\;$$
(8)$$\theta =\arccos\left[\frac{\left(x_{2}-x_{1}\right)\left(x_{3}-x_{1}\right)+\left(y_{2}-y_{1}\right)\left(y_{3}-y_{1}\right)}{\sqrt{\left[{\left(x_{3}-x_{1}\right)}^{2}+{\left(y_{3}-y_{1}\right)}^{2}\right]\left[{\left(x_{2}-x_{1}\right)}^{2}+{\left(y_{2}-y_{1}\right)}^{2}\right]}}\right]$$
(9)$$d=d_{E}+\alpha \left(\frac{\pi }{2}-\left|\theta -\frac{\pi }{2}\right|\right)d_{E}$$

Equation (7) presents the Euclidean distance formula, Equation (8) presents the formula for the assessment of the angle between the head of the vehicle and the specified stack angle, and Equation (9) presents the distance measurement function. $\left(x_{1},y_{1}\right)$ are the coordinates of the bulldozer; $\left(x_{2},y_{2}\right)$are the coordinates of the specified starting cell;$\;\left(x_{3},y_{3}\right)$ are the coordinates of the cell pointed at by the head of the bulldozer; $d_{E}$ is the Euclidean distance in metres from the bulldozer to the designated pile; θ is the angle between the course of the bulldozer and the specified starting cell; α is the angle weight; and d is the composite distance in metres.

In the process of guiding the bulldozer to the starting cell, as there are no obstacles in the work block, the bulldozer can only move in a straight line to the starting cell. When it travels to the starting cell, the move state changes to the work state, as shown in Figure 7.

### 4.3. Parameter Acquisition Based on Real-Time Monitoring

The OODA framework coupled with the CLA is required to be driven by the paving process parameters and the initial information of the dam storehouse surface. Based on the functionality of the CLA introduced in Section 4.1.2, the CLA can convert the real-time position information of bulldozers and the unloading information of dump trucks into the dynamic paving quality information of the storehouse surface. Thus, the following method is used to complete the collection of these two types of information. Paving process parameters are also the real-time construction information that is required to be obtained during the construction. The architecture of the real-time acquisition method for the paving operation parameters is presented in Figure 8.

To provide accurate and dynamic paving-quality information, it is necessary to accurately locate the real-time position of the bulldozer with the comprehensive use of GNSS technology, general packet radio service technology, and real-time kinematic technology; establish a differential base station in the construction area; and install a global positioning system (GPS) receiver, controller, and other monitoring equipment on the bulldozer [2]. The bulldozer can obtain the satellite positioning in real time and also perform real-time differential positioning on the base station by acquiring the positioning signal of the base station, thereby improving the positioning accuracy and timeliness [37].

It is necessary to obtain the discharge time and location to understand the dynamic pavement quality situation [53]. This can be realised by installing a BeiDou satellite positioning terminal and discharge sensor on the dump truck. The discharge sensor can sense the rise of the hopper and send the discharge signal to the BeiDou positioning terminal. The BeiDou positioning terminal can edit the position information and time information at this moment into a short message and send it to the server, which thereby obtains the discharge-related information.

## 5. Engineering Applications

The method proposed in this study was applied to a high earth-rock dam in southwest China. The dam is a core-wall earth-rock dam having a height of 295.0 m, total fill volume of 41.6 million m^3^, and a core wall area of 348 m length and 55.6 m average width. The field experiment of this study was conducted in the core wall area, and the bulldozer model used was SD32, manufactured by Shantui Construction Machinery Co., Ltd., Jining, China. A tablet equipped with an intelligent guidance program for the paving process was installed on the front windscreen of the bulldozer, such that the operator could see the arrow indication without any obstruction to the view from the windscreen.

### 5.1. Real-Time Acquisition Process of Paving Operation Parameters

As shown in Figure 9, using GPS + RTK technology, the monitor terminal is installed on the bulldozer, by receiving BeiDou, GPS, GLONASS satellite positioning system, and the difference back to the base station of the differential positioning data coupling calculation of the bulldozer real-time location data, the error location information within 2 cm, to meet the needs of paving process control [54].

This data can be represented by the following collection:(10)$$D_{bul}=\left\{x,y,H,v,t\right\}$$
where $D_{bul}\;$ is the real-time spatial position data of the bulldozer; x is the longitude; y is the latitude; H is the elevation in meters; v is the instantaneous speed of the bulldozer in m/s, and t is the time for obtaining the positioning in seconds.

By installing the BeiDou positioning device and unloading sensor on the dump truck, the unloading process can be sensed, and the positioning data can be sent back to the server through the conductor to achieve the acquisition and transmission of the unloading position information of the dump truck. The dump truck location information can be represented by the following set:(11)$$D_{dump}=\left\{x,y,H,t,S\right\}$$
where $D_{dump}$ is the location information of the dump truck, and S is the state of the dump truck (0 is the loaded condition, and 1 is the unloaded state).

The initial storehouse surface information should be determined by the field management personnel before construction begins, and the coordinates of the boundary points of the storehouse surface are determined using a measuring rod. The smoothness and average elevation of the pre-rolling process are then obtained according to the real-time monitoring system of the rolling quality [2] in order to obtain the position, size, smoothness, and other information of the storehouse surface of the dam before the construction.

In this study, to verify the superiority of the proposed framework, six sets of experimental data were constructed. First, three operators were each made to complete one paving operation, and the corresponding data were collected from the paving real-time monitoring system using these three sets of data as a comparison group. Then, one of the operators completed the work according to the instructions of the arrows; three experiments were conducted, and the data thus obtained were used as the experimental group. Finally, the six sets of data obtained were processed and analysed.

### 5.2. Dynamic Assessment

Dynamic assessment is applied to the action module of the OODA framework to evaluate the paving quality of the entire area. Therefore, the effect of applying dynamic assessments in engineering applications are also presented in a dynamic form.

The dynamic assessment comprises the evaluation of the paving quality at all times during the construction process. The Number 2 Test experiment set is considered as an example. Three stages are selected for the assessment: 10 min after the start of the paving, 25 min after the start of the paving, and after the completion of the paving. A three-dimensional colour map that reflects the elevation of the entire working area can be created based on the elevation information stored by each cell in the CA, as shown in Figure 10.

The various stages of the bulldozer operation can be clearly distinguished in the figure. In the beginning, as there are many unpaved places, it can be clearly observed that the paved and unpaved layers are connected in a stepped manner. Furthermore, owing to the existence of a mound in the area, red protrusions can be clearly observed in the colour map. At this time, the average paving thickness is generally below the standard value; thus, there is no need to consider the flatness. After approximately 25 min, similar features can still be observed, but the area of the paved portion becomes significantly larger, and the unpaved portion decreases. At this time, the average thickness may have reached the standard value, but it can be intuitively determined that the flatness has not yet met the requirement. When the paving is completed, it can be observed that the entire area is yellow in the map, and the elevation difference is small, indicating that the paving has been completed. This is when the average thickness and flatness have met the requirements.

In addition, the effect of the dynamic assessment can be compared horizontally with the effect of traditional work methods. The Number 3 Test experiment set and Number 3 Contrast comparison set are considered as examples for the analysis. The two sets are similar in size, shape, and elevation. The three-dimensional surface colour map can be plotted according to the elevation at each position at the end of the paving of the two sets, as shown in Figure 11. It can be clearly observed that the comparison set exhibits a serious ultra-thin phenomenon, while the thickness of each area of the experimental set is relatively average; thus, good results were obtained in the dynamic assessment [55].

### 5.3. Feedback Control

The key point of the dynamic control method is the path planning algorithm, and with the realisation of dynamic control, it is also necessary to take into consideration the method for outputting the algorithm result. As shown in Figure 12, in the application process, an intelligent guidance program for the paving process was compiled according to the above method to guide the travel of the bulldozer. The program is installed on the navigation tablet, and the tablet is installed on the windscreen. To avoid affecting the operator’s field of vision, augmented reality technology is used for the visual processing [56]. A real-time photograph obtained using the camera installed at the front of the bulldozer is used as a base map, and the path-planning result is displayed in the form of an arrow in the centre of the photograph. Thumbnails and real-time status information of the bulldozer are placed at other positions on the screen to allow the operator to understand the current situation.

A rudimentary Paving Quality Feedback Control algorithm (Algorithm 1) is outlined as follows:**Algorithm****1.****Paving Quality Feedback Control**(1) **Initialisation:****Input**: target area coordinates, initial mounds, start_height, and target_height(2) **Main cycle****Parameters:****Mound**-the number of soil mounds in the dump area of the dump truck **Endmain**-the number of iterations **Mounds.Add**-the additive operator of the number of soil mounds **Bulldozer.location**-bulldozer coordinate variables; **Get_location**-position sensing operator **Cell_Judge**-LA judges behaviour according to the state of neighbouring cells **PathPlanning**-path planning operator **Arrow**-optimal path indicator variable **Judge_quality**-bulldozer paving quality evaluation operator1. Endmain = 02. **While** Endmain == 0 Do3.    **IF** a new mound arises then4.   mounds.Add(new mound)5.   **End**6.   Bulldozer.location = Get_location()7.   Cell_Judge = get_arround(Bulldozer)8.   Arrow = PathPlanning(mounds, Bulldozer, Cell_Judge)9.   Show the Arrow on the Screen10. Update Bulldozer ‘s location and the cell that it passed just now11. Endmain = Judge_quality()12. **End**

In the algorithm pseudo code, the initial work block coordinates, initial mound information, and initial and target elevation of the block are entered during the initialisation phase. ***mounds.Add***(new mound) adds the information of the newly entered mound. ***Bulldozer.location*** = get_location() obtains the spatial coordinate information of the bulldozer. Cell_Judge = get_arround(Bulldozer) obtains the information of the cells around the bulldozer and stores it in Cell_Judge. Arrow = PathPlanning(mounds, bulldozer, Cell_Judge) indicates that the path planning algorithm analyses the information of the mound, bulldozer, and surrounding cells to obtain the current optimal path and presents it as an arrow. Endmain = Judge_quality() indicates that the paving quality (thickness and flatness) of the entire work block is analysed to determine whether the paving is completed. If it is completed, endmain becomes 1.

### 5.4. Discussion

The experimental data are listed in Table 1. The efficiency of the bulldozer in this study is defined based on the volume of gravelly soil paved by the bulldozer per unit time. The formula for calculating the efficiency is shown in Equation (12). The area in the table represents the total area of the work block; the amount of paved gravelly soil is converted according to the actual paving thickness and area (according to the site construction requirements, the thickness is generally 0.27 m); and the effective time is the time used for the paving operations (the time after neglecting the waiting time for parking, avoidance caused by the dump truck unloading, etc.). Owing to the influence of the construction progress on site, the size and shape of the work block cannot be guaranteed every time, and thus, the standard deviation of the elevation in each cell is used as an indicator for evaluating the flatness of the work block.
(12)$$E_{ffi}=\frac{S\cdot \bar{d}}{t}$$
where$\;E_{ffi}$ is the bulldozer efficiency, m^3^/s; S is the total area of the pavingsite, m^2^; $\bar{d}$ is average paving thickness, m; t is the effective time for paving operations, s.

The average value of the standard deviation of the comparison group is 0.095, and that of the experimental group is 0.0746, which is 21.5% less than that of the comparison group. In addition, from the longitudinal comparison within the group, the maximum value of the comparison group is found to be 0.1401, the minimum value is 0.0657, and the former is more than twice that of the latter. The maximum value of the experimental group is 0.0764, which is less than 6% greater than the minimum value of 0.0724. It can be observed that the use of this quality control method can stabilise and improve the quality of paving construction. The efficiency and flatness results of each group of experiments are shown in Figure 13.

Only by analysing the flatness of the standard deviation can we determine the thickness difference of each area in the entire work block, and this method is not sufficiently intuitive. Therefore, the elevation information of each cell at the end of the paving can be collected and expressed using different colours according to the difference in elevation, which comprises a graphical report. The graphical report can be used as another indicator for evaluating the flatness of the entire working block. Figure 14 presents the trajectory of each set and its corresponding graphical report, wherein the darker the colour, the greater the elevation, and the lighter the colour, the lower the elevation. The graphical reports of the comparison group are analysed. It can be observed that, except for the Number 2 Contrast comparison set, there exist obvious areas with ultra-thickness or ultra-thinness in the other sets. This indicates that uneven spreading of the material in the control group is common. Therefore, this system can effectively improve the construction quality and prevent the occurrence of uneven paving.

On comparing the trajectory maps of the two groups, it can be observed from Figure 14 that the trajectory maps of the experimental group are relatively uniform and dense. This is because the dynamic control method sets a strategy for levelling a mound and then leaving, such that the location and other information of the mound can be approximately inferred according to the density of the track. In addition, the ends of the trajectory of the experimental group are bent downward or upward. This is because the logic of the forward and backward movement in the work state, as designed in the program, is based on the elevation change of the GPS data, such that the bulldozer is vertically angled during downhill and uphill movements. Moreover, as the paving is performed along the horizontal axis, the majority of the intersection points are intersections of the ‘move’ path and ‘work’ path, and the moving path is primarily straight. It can be observed that the bulldozers of the experimental group work and move according to the path planning instructions.

In addition, the quality control method performs well in terms of efficiency, in the study, the volume of gravelly soil paved by paver per second is used to characterise the work efficiency. The minimum efficiency of the comparison group is 0.073 m^3^/s, and the highest efficiency is 0.137 m^3^/s. The efficiency of the skilled operators is nearly twice that of the novices. The gap between the experimental groups is significantly smaller: the highest efficiency is 0.127 m^3^/s, while the lowest is 0.109 m^3^/s. The difference between the maximum and minimum values is less than one-sixth. Therefore, the method proposed in this study can maintain its efficiency at the upper-middle level. Under the premise of ensuring construction efficiency, this method could play a role in improving and stabilising construction quality.

## 6. Conclusions and Future Research

Paving quality control is of great significance for ensuring the construction quality of a gravelly soil core-wall earth-rock dam. In this study, a method for the dynamic assessment and control of paving quality is proposed, and the following are the main results obtained:

The OODA framework coupled with the CLA is established to realise the dynamic assessment and control of the paving quality. The CLA improves the observe and orient modules in the OODA framework. The former converts the initial paving information into quality information through CAs and performs partition storage and partition updates. The latter interacts with the surrounding environment via LAs, such that the cells can be processed more specifically according to the mechanical operation.A dynamic path planning method for optimising the paving quality indicators is proposed, and this method is embedded in the decision module for realising intelligent guidance and control. The conducted experiments demonstrate that this method effectively reduces the dependence of the paving operations on manual experience and establishes a high-precision event feed control method, which improves the quality of the paving and stabilises the construction efficiency at a high level.The dynamic assessment method is embedded in the action module of the OODA framework for dynamically evaluating the paving quality information of the entire area updated in real time, which improves the comprehensiveness and timeliness of the assessment. The experiments demonstrate that this dynamic assessment method can be used to comprehensively and effectively evaluate the paving quality during the construction process and provide guidance for quality control.

The dynamic quality assessment and control method adopted in this study not only monitors and evaluates the quality situation during the paving process, but also realises quality control via paving-path guidance. This feedback framework of the coupled assessment and control has a certain reference significance for research in the field of hydraulic engineering. In addition, because this method can significantly reduce the reliance on manual experience in such applications, it introduces the possibility of realising unmanned paving operations [57].

## Figures and Tables

**Figure 1 sensors-21-07756-f001:**
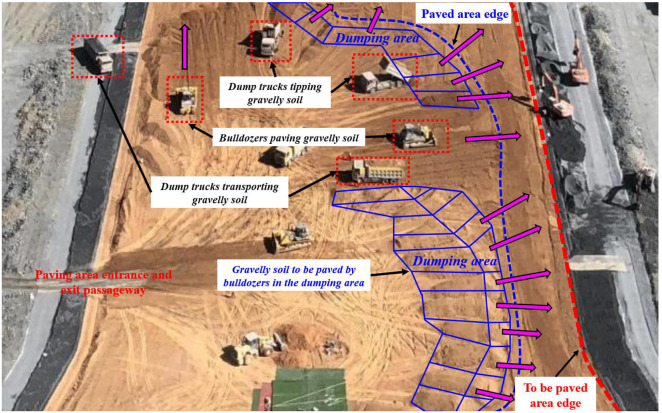
Paving construction process scene in core wall area of earth-rock dam.

**Figure 2 sensors-21-07756-f002:**
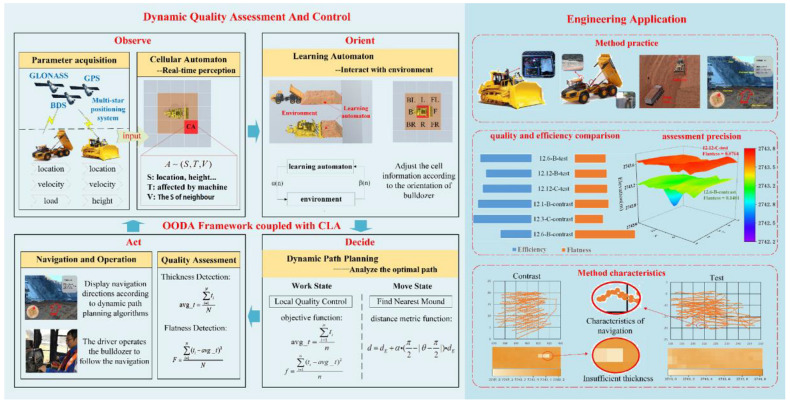
Composition and application of the proposed assessment and feedback control framework.

**Figure 3 sensors-21-07756-f003:**
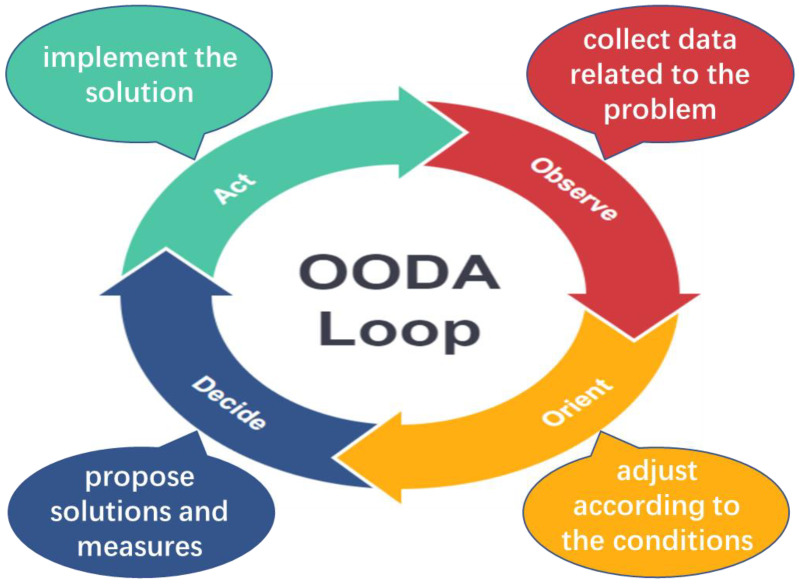
Theoretical framework of OODA loop.

**Figure 4 sensors-21-07756-f004:**
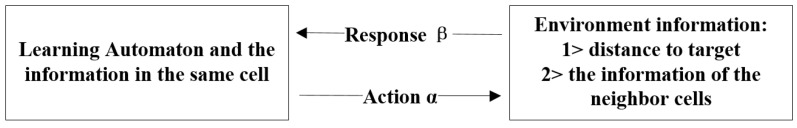
Relationship between the LA and the environment.

**Figure 5 sensors-21-07756-f005:**
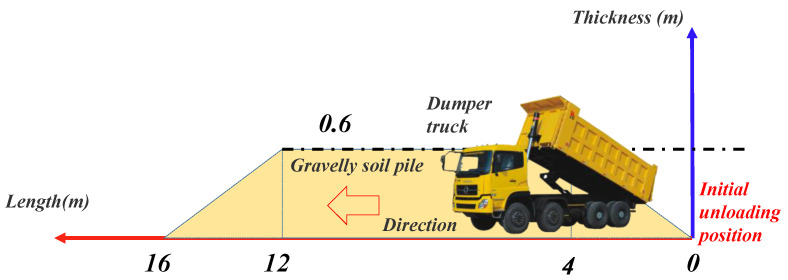
Unloading process diagram of dump truck.

**Figure 6 sensors-21-07756-f006:**
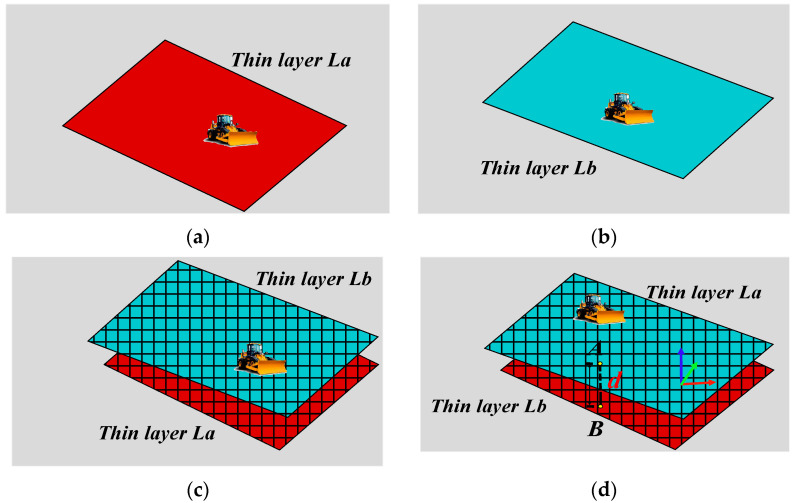
Calculation process of paving thickness of storehouse surface: (**a**) elevation of the previous storehouse; (**b**) elevation of storehouse in construction; (**c**) superimposition of (**a**,**b**); (**d**) calculation of paving thickness of storehouse surface.

**Figure 7 sensors-21-07756-f007:**
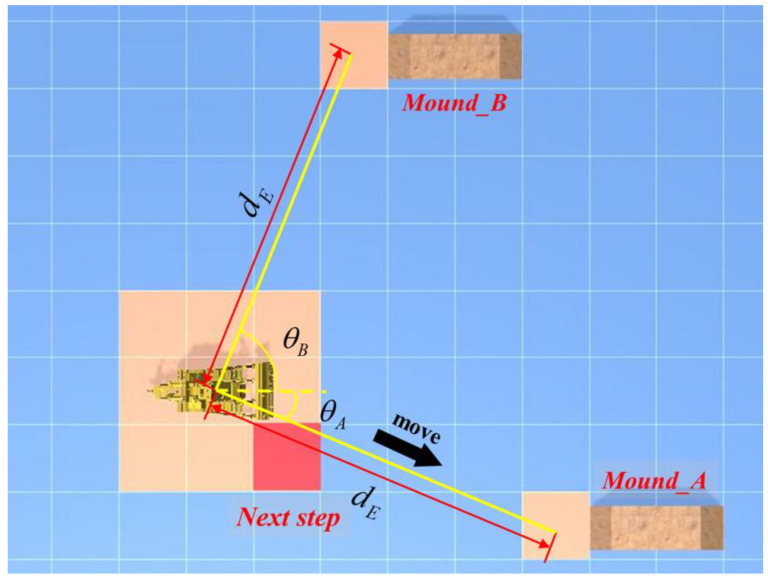
Feedback control of bulldozer by OODA decision module.

**Figure 8 sensors-21-07756-f008:**
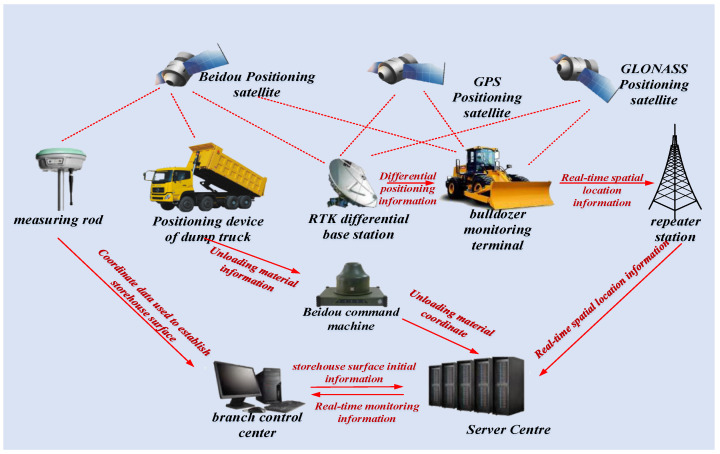
Real-time acquisition process of paving operation parameters.

**Figure 9 sensors-21-07756-f009:**
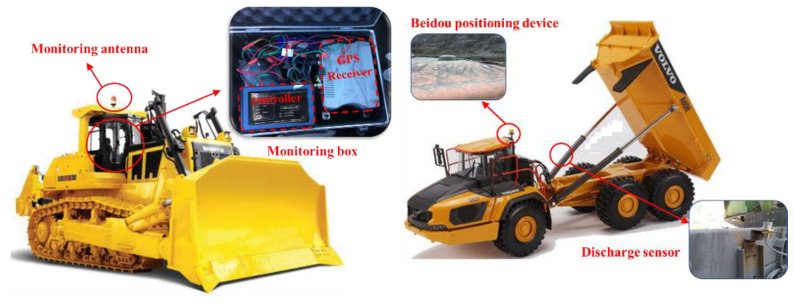
Equipment arrangement of monitoring terminal.

**Figure 10 sensors-21-07756-f010:**
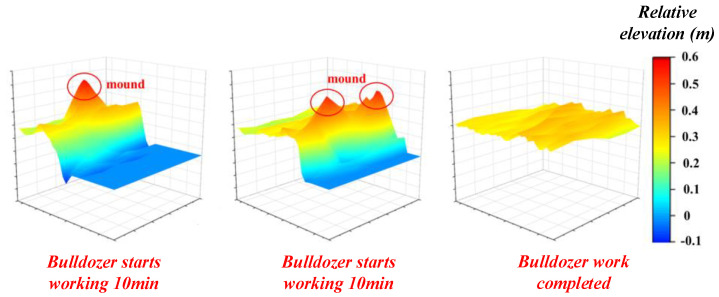
Three-dimensional colour map of the elevation of the entire working area.

**Figure 11 sensors-21-07756-f011:**
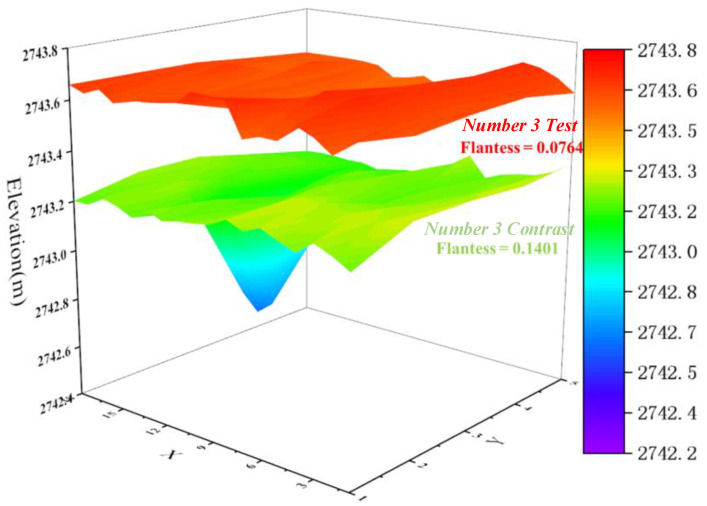
Three-dimensional surface colour map of the elevation at each position at the end of the paving of the two sets.

**Figure 12 sensors-21-07756-f012:**
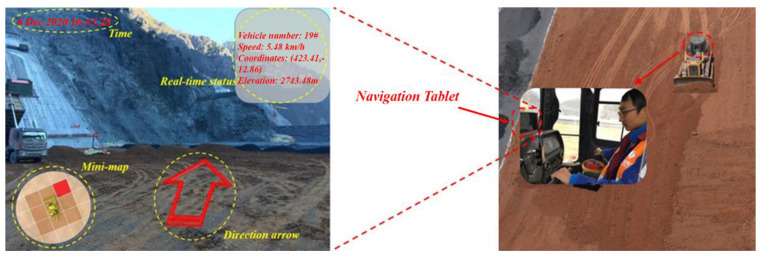
Real-time path optimisation based on feedback control.

**Figure 13 sensors-21-07756-f013:**
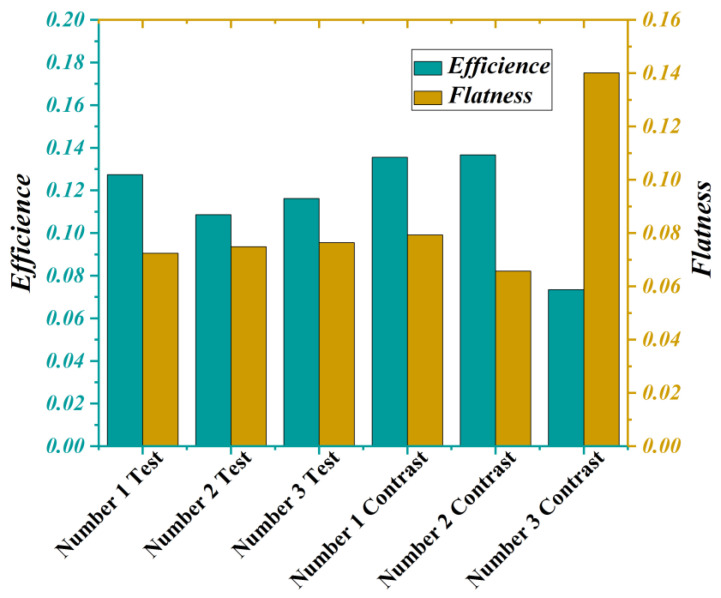
Comparative analysis results of paving efficiency and flatness.

**Figure 14 sensors-21-07756-f014:**
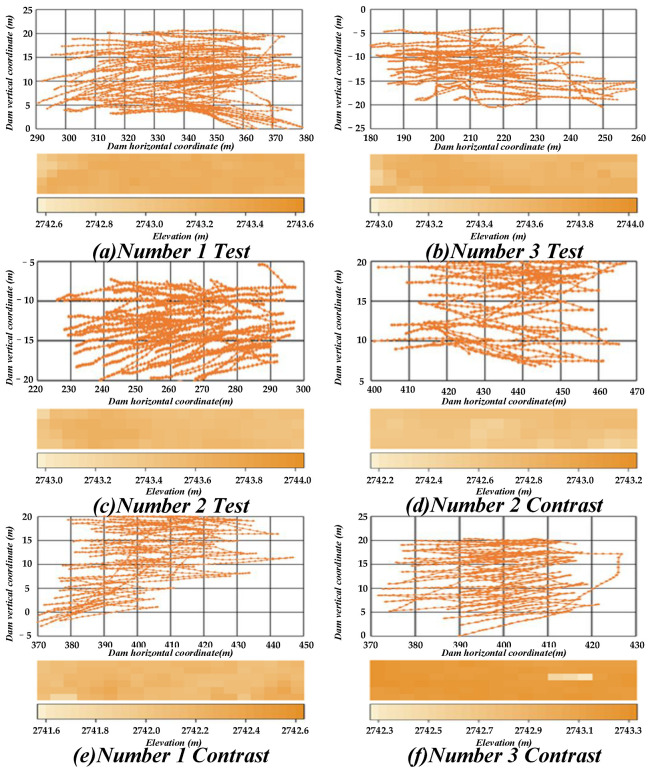
Trajectory of bulldozers and corresponding graphic reports for each test and contrast set.

**Table 1 sensors-21-07756-t001:** Experimental results.

Type	Name	Area (m^2^)	Time (s)	Efficiency (m^3^/s)	Average Efficiency (m^3^/s)	Flatness	Average Flatness
Experimental group	Number 1 Test	1360	2989	0.1274	0.1174	0.0724	0.0746
	Number 2 Test	845	2178	0.1086		0.0749	
	Number 3 Test	960	2314	0.1162		0.0764	
Contrast group	Number 1 Contrast	1044	2081	0.1355	0.1152	0.0793	0.095
	Number 2 Contrast	666	1315	0.1367		0.0657	
	Number 3 Contrast	891	3276	0.0734		0.1401	

## Data Availability

The data presented in this study are available on request from the corresponding author.

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
