# Peer review of "Assessment and Feedback Control of Paving Quality of Earth-Rock Dam Based on OODA Loop"

_sensors, 2021, doi:10.3390/s21227756_

Round 1
Reviewer 1 Report
- Dot missing after 'schedule' in the third line of Introduction, Please modify and check English editing of other chapters.
- Please modify the article structure and format according to the template.
- In the sixth line of Chapter 4.1.3, it is mentioned that cell stores the thickness. Is thickness obtained by GNSS or derived from other information? Please describe the process in detail.
- Please describe the application of cellular learning automaton in detail. Including but not limited to the rules of updating state information
- Is it necessary to describe the work state of the bulldozer with pictures? I think pictures will make the process easier to understand.
- In the second line of Chapter 4.3, it is mentioned that The CLA can convert position information and the unloading information into the dynamic quality information. Please describe the conversion process in detail.
Author Response
Detailed responses to Reviewers comments
Manuscript ID: sensors-1448048
Title: Assessment and feedback control of paving quality of earth-rock dam based on OODA loop
Type of manuscript: Article
Authors: Cheng Wang, Jiajun Wang *, Wenlong Chen, Jia Yu, Zheng Jiao,
Hongling Yu
Dear editor and reviewer,
Thank you for your letter and comments concerning our manuscript entitled “Assessment and feedback control of paving quality of earth-rock dam based on OODA loop” (Manuscript ID: sensors-1448048). Those comments are all valuable and very helpful for revising and improving our paper, as well as the important guiding significance to our researches. We tried our best to improve the manuscript and made some changes in the manuscript. The detailed responses to comments are shown as follows: The contents marked in blue represent reviewers’ suggestions and comments, contents marked in red are the revised portion in the revised paper, the contents marked in red with strikethrough represent the portion in the manuscript that has been deleted in the revised paper, and the black blot fonts represents the original portion which is kept in the revised paper.
We appreciate for Editor and Reviewers’ warm work earnestly, and hope that the correction will meet with approval.
Responses to Reviewer 1's comments
Reviewer #1:
Comment 1:Dot missing after 'schedule' in the third line of Introduction, Please modify and check English editing of other chapters.
Response 1:Thank you very much for your comments. We invested a lot of time and effort in adjusting and revising the content and grammar of the full paper. In addition, the revised paper was professionally linguistically touched up on the MDPI website. In the third line of the introduction, the dot after 'Schedule' has been filled in, and the same problem has been corrected in the full text.
- The revision is marked in red color in the revised paper (It can be seen in the forth line of the introduction) , the revised passage is shown as follows. Other grammatical and punctuation changes are more frequent, please see the revised draft for details
The paving operation is an important part of the whole construction process of earth-rock dams dam [1]. Generally speaking, bulldozers and dump trucks constitute the construction machinery for the paving of earth-rock dams for to meeting the tight construction schedule.
Comment 2: Please modify the article structure and format according to the template.
Response 2:Thank you very much for your comments. I have used the Microsoft Word template of Sensors Journal to re-prepare my manuscript. The manuscript section of the journal must have sections such as introduction, materials and methods, results, discussion, and conclusion, all of which are encapsulated in the revised manuscript. In our manuscript, an extensive search of current relevant studies was conducted in order to elaborate the characteristics and innovation of the study, so the literature review section is included as a separate section for realted work. also, in order to explain that all the methods are intended to serve the framework of quality assessment and feedback control of paving operations based on the Observation-Orientation-Decision-Action (OODA) cycle proposed in this study. Therefore, the research framework is first presented as a separate section so that it is easier for the reader to understand the various modules and methods presented later.
Comment 3: In the sixth line of Chapter 4.1.3, it is mentioned that cell stores the thickness. Is thickness obtained by GNSS or derived from other information? Please describe the process in detail.
Response 3:Thank you very much for your suggestions and comments. We totally agree with your suggestion and comments that we should add the detailed process of obtaining the thickness information of the storehouse surface. Therefore, we have added the corresponding process illustration at the corresponding position in the text to facilitate the reader to understand the process of acquiring thickness based on GNSS in detail.
- The revision is marked in red color in the revised paper (It can be seen in line 2-18, page 10) , the revised passage is shown as follows.
In the observation observe module, according to the characteristics of the cellular automaton CA, the work block is divided into equal-sized square grids, where each grid corresponds to a cell, and the cell stores the location coordinates, elevation, thickness, and other information of the corresponding grid [47]. The position coordinates and elevation information of the grid can be obtained directly by the GNSS equipment installed on the bulldozers, and the thickness information is the elevation difference between the corresponding grids of the upper and lower thin layers paved by the bulldozers. As shown in Figure 6, the GNSS installed on the bulldozer records the position information in real time. After the paving of the storehouse surface is completed, the fitted elevation of the storehouse surface is obtained by extracting the paving trajectory information of the storehouse surface as shown in Fig. 6a. When starting construction on a new layer, the system fits and updates the elevation map by acquiring the trajectory of the bulldozer in real time as shown in Figure 6b. Figure 6c shows the superimposition of the elevation maps of two paving thin layers. The thickness d at an arbitrary location can be obtained by calculating the height difference between the two planes, as shown in Figure 6d.
|
(a) |
(b) |
|
(c) |
(d) |
Fig. 6 Calculation process of paving thickness of storehouse surface: (a) Elevation of the previous storehouse; (b) Elevation of storehouse in construction; (c) Superimposition of a and b; (d) Calculation of paving thickness of storehouse surface.
Comment 4: Please describe the application of cellular learning automaton in detail. Including but not limited to the rules of updating state information
Response 4:Thank you very much for your suggestions and comments. We totally agree with your suggestion and comments that we should describe the application of cellular learning automaton in detail. In the related work subsection, we reintroduce the advantages of cellular learning automaton and add (but are not limited to) relevant literature in the field of rules of updating state information, presenting the current state of application of cellular learning automaton in different fields.
- The revision is marked in red color in the revised paper (It can be seen in line 25-43, page 4) , the revised passage is shown as follows. Due to the change of references, the serial numbers of the references in the full text have been corrected, please refer to the revised manuscript for details of the changes.
However, all mechanical actions in the paving process are construction quality-oriented, and merely relying on the monitoring of machinery is not enough insufficient for ensuring to reflect the quality of the whole entire warehouse. Cellular automata(CAs) has have the characteristics of regular state transitions transition and discrete update storage. In addition, the precise real-time positions position and grid partitions partition are essential for calculating and storing lift thickness information[11]. CLA, which is a hybrid of cellular automata (CA) and learning automata (LA), outperforms CA in terms of learning ability, making it more environment adaptable [13]. CLA is also preferable to LA because significant amounts of CA can induce more complicated phenomena than LA [20]. CLA integrates the benefits of CA in that the phenomena are mimicked by intelligible behavioral norms mixed with the learning capabilities of LA, rather than coupled functions [19], making computer simulations easier to execute. Since the introduction of CLA [21], this model and its extensions have proven their performance in solving problems, such as gas diffusion [16], land use evolution [17], urban development [18] and flood evolution[15] community detection [22], wireless sensor network [23], resource allocation [24], and emergency evacuation management for updating pedestrian location status information updates. So far, CLA has been mainly used to solve optimization problems and has rarely been used to model the self-organizing behavior of automated equipment [12].
Comment 5: Is it necessary to describe the work state of the bulldozer with pictures? I think pictures will make the process easier to understand.
Response 5:Thank you very much for your suggestions and comments. In order to make it more convenient for the reader to understand the working state of the bulldozer, in Figure 2, we have visualized the working state of the bulldozer under each module of observe--orient--decide--act in the framework of OODA loop-based assessment and feedback control of paving quality of earth-rock dam. By showing the different operating states of the bulldozer in the same framework diagram, the quality evaluation and feedback control process of assessment and feedback control of paving quality of earth-rock dam in this paper can be easily understood. At the same time, in the subsequent subsections, additional graphics are added for additional explanation, for example, Figure 9 shows the real-time collection of positioning information on the dozer while the observe module is operating, and Figure 6 shows the real-time sensing of storehouse surface thickness and other information by the dozer based on the cellular automaton. To make the reader more aware of the working state of the dozer, the corresponding descriptions are added under Figure 2.
- The revision is marked in red color in the revised paper (It can be seen in line 17-28, page 6) , the revised passage is shown as follows.
The engineering application part is a demonstration of the effects and characteristics of thise method when applied to engineering applications. The Eexperiments and an analysis of their results show that this method provides a superior performance as compared this method is superior to manual work. The composition and application of this the framework are shown presented in Figure 2. The OODA framework consists of 4 modules: observe, orient, decide, and act. In the observe module, the dozer senses information about the entire paving area in real time based on a CA. In the orient module, the LA helps the dozer to interact with the environmental information to achieve the adjustment of the dozer. In the decide module, a dynamic path planning algorithm is used to plan the optimal path for the bulldozer in order to achieve the paving quality target adjustment. In the act module, the dozer is executed by navigation and the current paving quality is dynamically evaluated.
Comment 6: In the second line of Chapter 4.3, it is mentioned that The CLA can convert position information and the unloading information into the dynamic quality information. Please describe the conversion process in detail.
Response 6:Thank you very much for your suggestions and comments. The dynamic quality information mentioned in this paragraph refers to the dynamic paving quality information of the storehouse surface. The corresponding paragraphs have been corrected to avoid ambiguity. Since the focus of subsection 4.3 is to introduce the collection of real-time position information of bulldozers and the unloading information of dump trucks, the detailed conversion process of CLA to convert the real-time position information of bulldozers and the unloading information of dump trucks into the dynamic quality information of the work block is suitable to be introduced in section 4.1.2. In the second line of Chapter 4.3, to avoid ambiguity, we have added " Based on the functionality of the CLA introduced in section 4.1.2".
- The revision is marked in red color in the revised paper (It can be seen in line 15-17, page 13) , the revised passage is shown as follows.
The OODA framework coupled with the CLA is required to be driven by the paving process parameters and the initial information of the dam storehouse surface. Based on the functionality of the CLA introduced in section 4.1.2,The the CLA can convert the real-time position information of bulldozers and the unloading information of dump trucks into the dynamic paving quality information of the work block storehouse surface .
- The revision is marked in red color in the revised paper (It can be seen in line 23-47, page 8 and line 1-26, page 9 ) , the revised passage is shown as follows.
CLA can convert the real-time position information of bulldozers and the unloading information of dump trucks into the dynamic paving quality information of the storehouse surface.
The storehouse surface can be divided into a 1m × 1m square grid, each grid corresponds to a cellular, and the location coordinate, elevation, thickness and other information of the corresponding grid are stored in the cellular, and each state information of the cellular corresponds to its state transition rule. the transfer of each state in the cellular is carried out in the following order:
1 The cellular coordinates will not change after the storehouse surface division stage is obtained.
2 The cellular activation value is determined by obtaining real-time monitoring data and the activation status information of the surrounding cells. The inactive state value is 0, when the bulldozer spatial position coordinates coincide with the cellular coordinates, the activation value is 1, when the activation value of the surrounding cell is 1, the activation state value of the cell becomes 1.1. When the spatial position coordinates of the dump truck coincide with the cellular coordinates , the activation value of the cell is changed to 2, and the activation values of the cellular coordinates and are changed to 2, thus determining that the activation values of the cells existing on the width of the mound are all updated to 2. In order to ensure that all the cells on the length of the mound are activated according to position, the activation value of cellular coordinates can be changed to 2.01, and so on. The other activation value of the cell is, then the activation value of the cells on both sides of the y direction is also changed to , The activation value of the cell at the position becomes ;
3 The elevation can be obtained in different ways according to the activation value of the cell, when the activation value is 1 or 2, the elevation is equal to the elevation value in the real-time monitoring data of the bulldozer or dump truck. When the activation value is 1.1, because the cell is under the bulldozer, the elevation value is equal to the elevation value of cell with activation value of 1 at this time. When the activation value is , each cell corresponds to each position of the unloading pile. Elevation updates should be made according to Equation (3)
|
(3) |
Where, is the updated elevation of this cell,m; is the pre renewing elevation of this cell,m; is the distance from the starting unloading position as shown in Figure 5,m.
The thickness is generally updated according to the elevation value in the cell, and because the thickness update is the last step of the cell state update, the active state value is set to 0 after the thickness update. Thickness state transfer should satisfy the Equation (4).
|
(4) |
Where, d is the thickness,m; is the updated elevation of the cell,m;is the pre renewing elevation of this cell,m.
To sum up, the cellular automata model of gravelly soil paving quality is established, the storehouse surface is divided into grids, and the real-time monitoring data are obtained by using cellular automata. Then, according to the state transition rules, the paving quality information of each location is transformed and stored in the corresponding cell. As a result, the action of construction machinery is connected with the paving quality, which provides a high-precision and real-time data source for path planning.
Figure 5. Unloading process diagram of dump truck.
The above is our responses to your comments. Special thanks to you for your valuable comments.
Remark: The other modified parts of the revised paper are also marked in red, which include the added references, the added equation (Equation 3,4,12),the improved language, and added figure ( Fig 5,6).

Reviewer 2 Report
The paper is generally well written and structured. I liked both the idea and research plan of the study. But, authors need to thoroughly revise the manuscript to enhance data representation and writing parts. Given the following shortcomings, the manuscript requires minor revisions.
- There are many facts and statements are discussed in the manuscript without quoting the proper references. Some examples are - “Failure to meet these requirements may lead to unqualified paving results such as uneven paving, thus affecting the compactness quality of the final earth-rock dam and reducing the service life of the dam”, “An effective leveling quality control plays a pivotal role in increasing productivity in earth rock dam construction”, “When the leveling layer thickness is too thick, there is a serious separation of coarse and fine materials, which results in unstable and unsafe operation of earth-rock dams”, “At the same time, for earth-rock dam construction, it is difficult for an inexperienced manipulation to know whether a paving layer achieves the designed target, causing more work time and resources consumption” and “Most importantly conventional leveling quality control mainly depends on random spot test after the leveling work.”. Please thoroughly check the manuscript and provide appropriate references.
- Flow diagram – follow standard flow diagram symbols and rules.
- Please maintain the aspect ratio of the figures/images.
- Figure 8 and 10 – provide the reference of the figure in the running text.
- Figure 12 – please redraw the figure with constant width and Hight for all graphs.
- Provide x-label ad x-label in all the graphs (Figure 8 and 12).
- There are numerous errors of “Tense and Grammar” throughout the manuscript. Therefore, diligent editing is required to fix the ENGLISH language.
Reviewer 3 Report
This paper presented a method of paving quality dynamic assessment and control based on the observe-orient-decide-act (OODA) loop. This study has great engineering significance for paving quality control. But the structure and content of the article need to be further improved.
(1) In section 5.3
The two paragraph “On comparing the trajectory maps of the two groups, it can be found that the trajectory maps of the experimental group are relatively uniform and dense.…. Under the premise of ensuring construction efficiency, this method could play a role in improving and stabilizing the construction quality.” Should be in the section 5.4 Discussion.
(2) Each figure shall be cross referenced and explained in the text. Figure 10 did not appear in the text
(3) In section 5.4
The names of the experimental groups and the contrast groups should be easier to understand rather than directly using the project number.
(4) The definition and calculation formula of efficiency need to be supplemented.
(5) From the figure 11, the results had shown that using the method proposed in this paper, the stability was improved, but the efficiency and flatness were not improved very much. How to explain that?
(6) In figure 8 and figure 12, the coordinate axis needs to write the title and unit clearly.
Round 2
Reviewer 1 Report
Congratulations. The manuscript has been modified well to publish.
Reviewer 3 Report
The authors have responded to my questions one by one and revised the article. The article can be published.